# Two distinct superconducting states controlled by orientations of local wrinkles in LiFeAs

Lu Cao [1,2,5], Wenyao Liu [1,2,5], Geng Li [1,2,3,4,5 ✉], Guangyang Dai[1,2], Qi Zheng [1,2], Yuxin Wang[1,2], Kun Jiang[1,2], Shiyu Zhu [1,2], Li Huang[1,2,3], Lingyuan Kong [1], Fazhi Yang[1,2], Xiancheng Wang[1,3,4], Wu Zhou [2,3], Xiao Lin[2,1], Jiangping Hu[1,2,3], Changqing Jin [1,3,4], Hong Ding [1,3,4 ✉] & Hong-Jun Gao [1,2,3,4 ✉]

For iron-based superconductors, the phase diagrams under pressure or strain exhibit emergent phenomena between unconventional superconductivity and other electronic orders, varying in different systems. As a stoichiometric superconductor, LiFeAs has no structure phase transitions or entangled electronic states, which manifests an ideal platform to explore the pressure or strain effect on unconventional superconductivity. Here, we observe two types of superconducting states controlled by orientations of local wrinkles on the surface of LiFeAs. Using scanning tunneling microscopy/spectroscopy, we find type-I wrinkles enlarge the superconducting gaps and enhance the transition temperature, whereas type-II wrinkles significantly suppress the superconducting gaps. The vortices on wrinkles show a $C_2$ symmetry, indicating the strain effects on the wrinkles. By statistics, we find that the two types of wrinkles are categorized by their orientations. Our results demonstrate that the local strain effect with different directions can tune the superconducting order parameter of LiFeAs very differently, suggesting that the band shifting induced by directional pressure may play an important role in iron-based superconductivity.

[1] Institute of Physics, Chinese Academy of Sciences, Beijing 100190, China. [2] School of Physical Sciences, University of Chinese Academy of Sciences, Beijing 100049, China. [3] CAS Center for Excellence in Topological Quantum Computation, University of Chinese Academy of Sciences, Beijing 100190, China. [4] Songshan Lake Materials Laboratory, Dongguan, Guangdong 523808, China. [5]These authors contributed equally: Lu Cao, Wenyao Liu, Geng Li. ✉email: gengli.iop@iphy.ac.cn; dingh@iphy.ac.cn; hjgao@iphy.ac.cn

The origin of superconductivity in iron-based super-conductors (FeSCs) remains elusive despite intensive research efforts over a dozen years[1-3]. The large orbital degrees of freedom, as well as the presence of intertwining orders hinder a microscopic understanding of the pairing mechanism in FeSCs (refs. [4-6]). As a perturbation method, external pressure can lift the ground state degeneracy and offer detailed information about how the unconventional superconductivity evolve with other electronic orders. For example, an in-plane resistivity anisotropy[7] and a spin excitation[8] have been observed in electron-doped $BaFe_2As_2$ under uniaxial pressure; in FeSe with external pressure, the magnetic order could emerge and coexist with the high-temperature superconductivity[9], while enhanced spin fluctuation is evidenced as well[10]. Among FeSCs, LiFeAs is unique as its phase diagram is not intervened by any magnetic or nematic order, which provides an appropriate platform to detect the relationship between pressure and superconductivity in strong correlated system[11,12].

Under hydrostatic pressure, transport measurements reveal the transition temperature $T_c$ of LiFeAs declines linearly with increase of pressure amplitude, with no derived order, such as magnetism or nematicity observed[11]. However, when the uniaxial strain is applied to the specific lattice direction, stabilized sematic electronic state emerges, and suppresses the paring strength of superconductivity[12]. These divergent reports demonstrate that, compared with hydrostatic pressure, the orientation-dependent strain would trigger quite different influences on LiFeAs. Given the existence of multiple Fermi surfaces in the Brillouin zone, as well as the physical regime of corelated interactions in this material, the superconducting order parameter affected by orientation-dependent strain are intricate. Therefore, more experimental observations are required.

In this work, by using scanning tunneling microscopy/spectroscopy (STM/S), we report the observation of orientation-dependent superconductivity at two types of wrinkles on LiFeAs surface. The tunneling spectra show an increase of super-conducting (SC) gaps on type-I wrinkles and a reduction on type-II wrinkles. Compared to wrinkle-free region, temperature-dependent measurements of the SC gap show that the gap-closing temperature on type-I wrinkle region is enhanced by 20–30%, but it remains almost unchanged on type-II wrinkle region. Although wrinkles are a commonly consequence of relieving transverse strain induced by the change of atom coordination[13,14], the spatial feature of superconducting vortices observed on wrinkles confirms the existence of local strains. While the wrinkle orientations are continuous, the associated SC gap size has an abrupt jump at a certain angle of orientation, which is likely the consequence of Lifshitz transition.

## Results

### The wrinkle topography on LiFeAs surface.
The atomic model of LiFeAs is shown in Fig. 1a. Unlike many other FeSCs (refs. [15,16]), the cleavage of LiFeAs crystal occurs at a nonpolar plane between the two Li layers (dashed line in Fig. 1a), pre-senting a good platform for investigating and tuning the unconventional superconductivity at the nanoscale[17-20]. The stoichiometric LiFeAs shows superconductivity below the tran-sition temperature $T_c$ (~17 K)[21]. Two types of wrinkles are observed on LiFeAs surface (Fig. 1b–d), appearing as straight 1D ridges. Type-I wrinkles extend along the [110] direction (with respect to Li surface, also the Fe-Fe direction) or its neighboring directions, spanning a width about 15 nm with a maximum height of ~1.0 Å (lower panel, Fig. 1c). Type-II wrinkles extend along the [100] direction (also the Fe-As direction) or its neighboring directions, with a width of ~10 nm and a maximum

height of ~0.7 Å (lower panel, Fig. 1d). Both types of wrinkles are uniform in width and extend from several tens of to hundreds of nanometers. An atomic resolution image of type-I wrinkle in Fig. 1e shows a continuous and perfect Li lattice, excluding the possibility of formation of twin boundary[22,23], domain wall[24] or line defects[25]. Within the resolution of STM, no obvious lattice constant change can be detected (Supplemental Fig. 1). We propose that these wrinkles are likely induced by releasing of local strain during the creation of LiFeAs surface upon cleavage. Indeed, by atomic force microscopy (AFM), we explicitly demonstrate that the wrinkles on LiFeAs surface have real spatial corrugations, instead of purely reflecting as enhancement of local density of states (LDOS) (Supplemental Fig. 2).

### Local superconductivity tuned by two types of wrinkles.
There is remarkable differences between the type-I and II wrinkles in their LDOS. Figure 1f displays the differential conductance spectra ($dI/dV$) taken at the wrinkle-free region (black curve), type-I (red curve) and type-II wrinkles (blue curve) on the sur-face, as marked by the crosses in Fig. 1c, d. In the wrinkle-free region, $dI/dV$ spectrum shows multigap features of LiFeAs, with a large gap of ~5.8 meV (possibly come from the inner-hole pocket at the Γ point) and a small gap of ~2.9 meV (possibly come from the outer-hole pocket at the Γ point), which are consistent with previous reports[17-19,26]. Intriguingly, the tunneling spectrum on type-I wrinkle yields a coherence peak at 7.3 meV and a shoulder at 3.6 meV, and type-II wrinkle exhibits a single V-shaped gap of 2.5 meV. We note that electron doping[20,27] and application of external pressure[11,12] normally lead to reduction of SC gap or $T_c$, thus the increase of SC gaps is not commonly observed in LiFeAs.

Inspired by the novel gap features observed on the wrinkles, we carry out spatial $dI/dV$ spectra line-cut across and along the wrinkles, as displayed in Fig. 2a–d. On a type-I wrinkle, the coherence peaks of $\Delta_1$ start to shift to higher energies when getting close to the wrinkle edge and show the constant values of $\pm7.3$ meV across the wrinkle (Fig. 2a). At the same time, the shoulders of $\Delta_2$ follow a similar tendency. The enlarged gaps $\Delta_1$ and $\Delta_2$ remain homogeneous along the type-I wrinkle (Fig. 2b). Furthermore, the gap map of a type-I wrinkle (Supplemental Fig. 3a, b) reveals the maximum gap size ($\Delta_1$) distribution. It is evident that the type-I wrinkle has larger SC gap sizes compared with the wrinkle-free region. Also, the edges of the wrinkle (Supplemental Fig. 3a, b) show the largest SC gap, possibly suggesting the highest strain at the edges. We compare the SC gaps between the wrinkle-free region and type-I region under different axis scales (Supplemental Fig. 3c). The results show that the enhancement of the two SC gaps is of the same ratio (20–30%). On a type-II wrinkle, however, the coherence peaks of $\Delta_1$ are strongly suppressed when getting close to the wrinkle edge and totally disappear on the wrinkle region. Meanwhile, the shoulders of $\Delta_2$ evolve into a pair of coherence peaks on the wrinkle (Fig. 2c). Note that the gap features are robust and homogenous along the type-II wrinkle (Fig. 2d).

We next perform the temperature-dependent $dI/dV$ measure-ments. On a type-I wrinkle, the SC gap $\Delta_1$ can be well differentiated at 17 K ($T_c$ of bulk LiFeAs) and gradually closes at ~20.5 K (Fig. 3a), while the gap closes at 17 K at the wrinkle-free region (Fig. 3b). The SC gap of a type-II wrinkle also closes at ~17 K (Fig. 3c). We note that there is a bump near $E_F$ at high temperature (black arrows in Fig. 3a, b), which would be the band top of $d_{xz}$ (ref. [28]). In Fig. 3d, we plot the extracted gap values as a function of temperature (Supplemental note 11). The wrinkle-free region (black squares) and type-I region (red triangles) follow the same tendency, which is more obvious after rescaling the data of type-I region (pink triangles). This coincidence implies that the

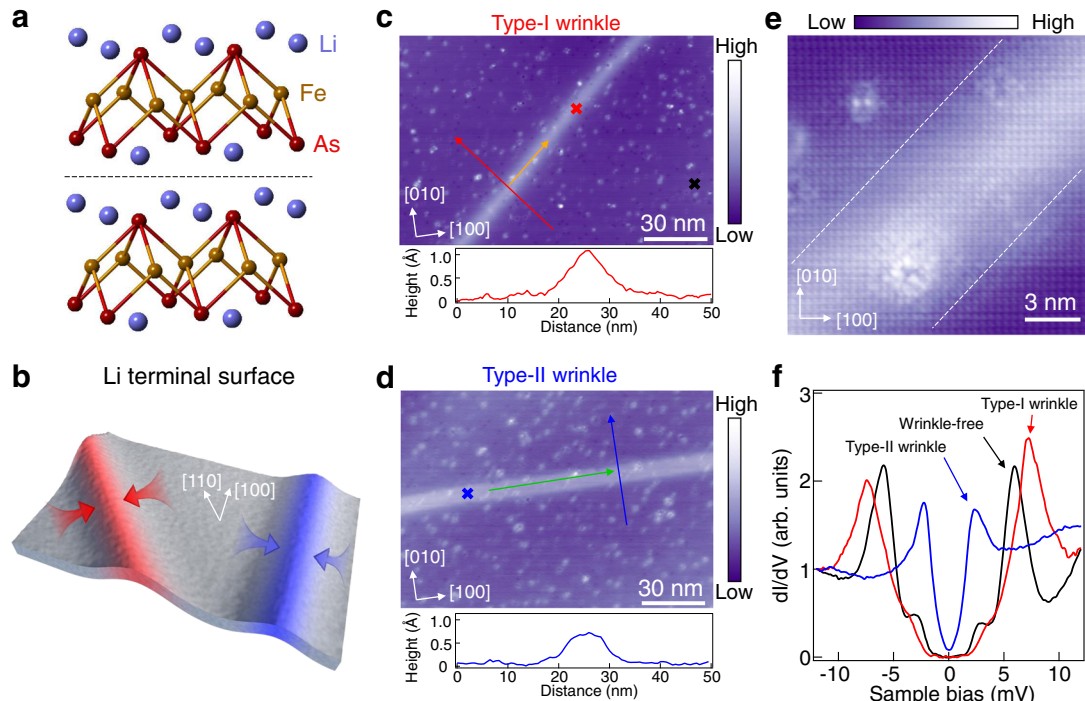

**Fig. 1 Strain-induced wrinkles on LiFeAs surface. a** The crystal structure of LiFeAs. The black dashed line indicates the location where the cleavage happens. **b** The sketches of the two types of wrinkles and the local strain on LiFeAs surface. **c** Upper panel: a large scale STM topography of LiFeAs surface showing a type-I wrinkle. The [100] and [010] mark the lattice directions on the Li terminal surface, respectively. Setpoint: $V_s = -20$ mV, $I_t = -20$ pA. Lower panel: a line profile taken along the red line in the upper panel. **d** Upper panel: a large scale STM topography of LiFeAs surface showing a type-II wrinkle. Setpoint: $V_s = -30$ mV, $I_t = -20$ pA. Lower panel: a line profile taken along the blue line in the upper panel. **e** The atomic resolution image of type-I wrinkle. The white dashed line indicates the wrinkle edge. Setpoint: $V_s = -3$ mV, $I_t = -1$ nA. **f** The $dI/dV$ spectra taken at the three crosses in (**c** and **d**). Compared with the SC gaps ($\Delta_1 = 5.8$ meV, $\Delta_2 = 2.9$ meV) detected at wrinkle-free region (black curve), two enlarged gaps ($\Delta_1 = 7.3$ meV, $\Delta_2 = 3.6$ meV) is observed on type-I wrinkle (red curve), while a smaller gap (2.5 meV) is observed on type-II wrinkle (blue curve). Setpoint: $V_s = -10$ mV, $I_t = -200$ pA.

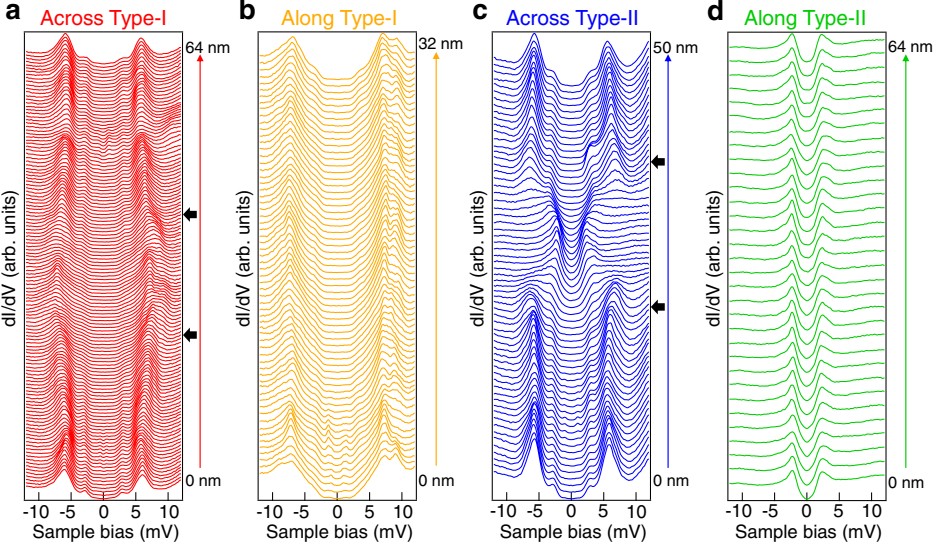

**Fig. 2 Superconducting gaps on type-I and type-II wrinkles. a**, **b** The waterfall plots of $dI/dV$ spectra across (red) and along (orange) the type-I wrinkle as marked by the arrows in Fig. 1c, showing the homogeneous and enlarged superconducting gap along type-I wrinkle. The black arrows in (**a**) indicate the type-I wrinkle boundary. **c**, **d** The waterfall plots of $dI/dV$ spectra across (blue) and along (green) the type-II wrinkle as marked by the arrows in Fig. 1d, showing the homogeneous and reduced superconducting gap along type-II wrinkle. The black arrows in (**c**) indicate the type-II wrinkle boundary. Setpoint for (**a–d**): $V_s = -10$ mV, $I_t = -200$ pA.

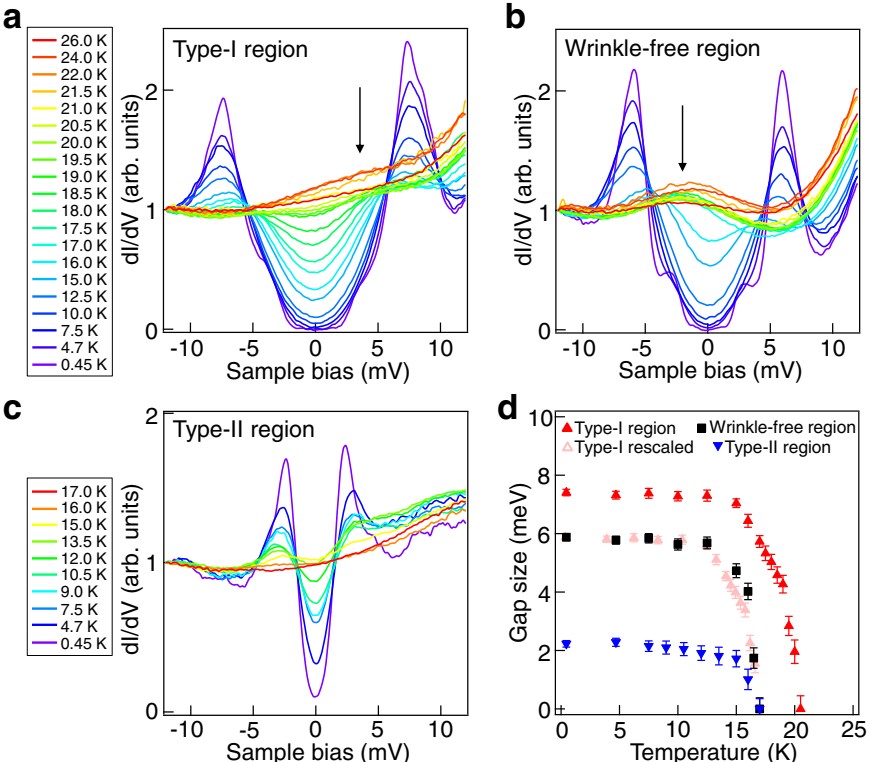

**Fig. 3 Temperature dependence of superconducting gaps on the wrinkles. a–c** Temperature-dependent d$I$/d$V$ spectra taken at type-I wrinkle region, wrinkle-free region and type-II wrinkle region, respectively. The black arrows highlight the bump features of the LDOS. (**a**) and (**b**) share the same legend. Setpoint: $V_s = -10$ mV, $I_t = -200$ pA. **d** A plot of the gap sizes as a function of temperature extracted from **a–c**.

type-I region has similar coupling strength with that of the wrinkle-free region. However, on the type-II region, super-conductivity behaves differently.

**The C$_4$ symmetry breaking and two distinct superconducting states on wrinkles**. To explain the exotic superconducting behaviors we observed, the first step is understanding the circumstances on these wrinkles. Here, we image the vortex structure by zero-bias conductance (ZBC) map (Supplemental Fig. 4). The vortex on the wrinkle-free region present a four-pointed-star like structure due to the C$_4$ symmetry of the Fermi surface of $d_{xy}$ band[29] (Fig. 4a). In contrast, the vortex at wrinkles change from C$_4$ symmetric to C$_2$ symmetric shape with the long axis extending along the wrinkle orientation (Fig. 4b). We point out that our vortex structure results not only clarify the existence of local strain on the wrinkles, but also suggests that the configuration of Fermi surface changes consequently.

Next, we plot the statistics of the orientation of the wrinkles and the corresponding SC gaps, which reveals that the [110] direction favors the type-I wrinkles while the [100] direction favors the type-II wrinkles (Supplemental Figs. 9 and 10). Intriguingly, wrinkles along other orientations are also found in our experiment (Fig. 4c, Supplemental Figs. 9 and 10), whereas the gap size versus orientation yields an abrupt change at about 20° with respect to the [110] direction. The statistical results indicate that the SC gaps do not gradually evolve with wrinkle orientations, but resemble a discontinuous transition (Fig. 4c). We also note that the wrinkles can make turns on the surface, leading to a transition from type-I to type-II (Supplemental Fig. 5).

Further analysis indicates that the switch of the super-conductivity on two-type wrinkles is coincidence with the band structure changing under different orientations of local strain.

First, the bump features in Fig. 3a, b indicate the band of $d_{xz}$ shifts above the Fermi level on the type-I wrinkles (black arrows in Fig. 3a, b). Next, our results indicate that the wrinkles impact the LDOS away from the Fermi level, as well (Fig. 4d). Note that there is a shoulder around 33 meV which has a slight shift comparing to the one in the wrinkle-free region (Supplemental Fig. 6), suggesting the band top of the $d_{xy}$ orbital shifts up at the type-I wrinkle and down at the type-II wrinkle[28,30]. In addition, the whole gap feature changes from U-shape to V-shape[20] with nonzero LDOS at the zero energy on the type-II wrinkle (Figs. 1f, 3c, and 4d), and the intensity of LDOS for the type-II wrinkle has a large loss comparing to others as well, both indicate a possibility that the inter-hole bands ($d_{xz}$ and $d_{yz}$) sink down below the Fermi level on the type-II region.

## Discussion

We note that in previous reports which utilize the external global pressure, only suppressions of superconductivity were found either in the hydrostatic pressure or uniaxial strain, while we simultaneously observed the enhancement and suppression of superconductivity on type-I and type-II wrinkles, respectively. In addition, our results reveal that the unique orientation-dependence of gap size separated by an abrupt jump (Fig. 4c), which may suggest the detailed responding mechanism to strain effect is diverse when the strain effect is shrink to nanoscale.

In the Supplemental Table 1, we summarize the super-conducting gap size, $T_c$ and the ratio $2\Delta/k_B T_c$ for wrinkle-free, type-I and type-II wrinkles. The ratios are similar for wrinkle-free and type-I wrinkles, indicating that the paring strength at type-I wrinkle region is close to that at the wrinkle-free region. Therefore, the increase of DOS at type-I wrinkle as proposed here is reasonable. On the other hand, $\Delta_2$ at type-II wrinkle is small and the ratio at type-II wrinkle (~3.4) is close to the weak-coupling

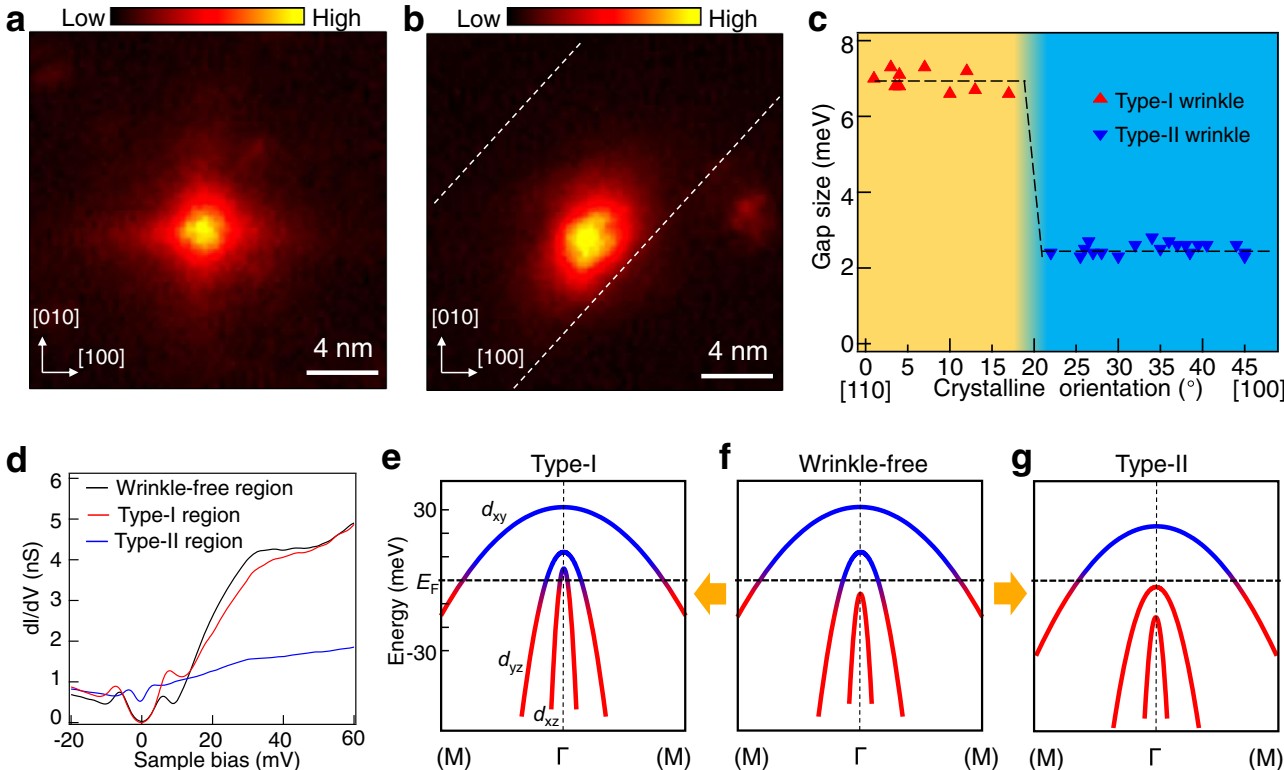

**Fig. 4 Uniaxial strain-induced band shifting. a, b** Zero-bias conductance map of vortices at wrinkle-free region and type-I wrinkle, respectively. The white dashed line in (**b**) indicate the wrinkle edge. Setpoint: $V_s = -10$ mV, $I_t = -200$ pA. **c** Statistics of orientations of the two types of wrinkles. The dashed line is a guide to eye. **d** The wide range of d$I$/d$V$ spectra. Setpoint: $V_s = -100$ mV, $I_t = -200$ pA. The unit for d$I$/d$V$ spectra is set to nS for the direct comparison of the LDOS at the three regions. **e-g** Sketches of the band structures of LiFeAs near the $\Gamma$ point and the band shifting of two types of wrinkles. The energy scale is an estimation, not an accurate value.

BCS regime, suggesting that the $T_c$ at type-II wrinkle may be induced by the bulk superconducting proximity effect. Another possible explanation is that the $d_{xz}$ as well as $d_{yz}$ bands sink below $E_F$, and only the smaller gap of $d_{xy}$ is observed (tunneling at the STM junction is insensitive to the M point), since the gap function in the FeSCs is found to be approximately proportional to $\cos(k_x)\cos(k_y)$[31]. In this way, $T_c$ is not driven by $\Delta_2$ which is much smaller than the global superconducting pairing strength.

Based on the observations discussed above, we propose a possible scenario that the local strain on wrinkle causes the shifting of bands around $\Gamma$ point (Fig. 4e–g). In bulk LiFeAs, the $d_{xz}/d_{yz}$ bands split possibly due to the spin-orbit coupling[28], with $d_{yz}$ crossing $E_F$ and $d_{xz}$ sinking below (Fig. 4f). When local strain exists along the [110] direction (Fe-Fe direction), the $d_{xz}$ band top shifts above $E_F$, giving rise to an increase of DOS near the $\Gamma$ point (Fig. 4e). The enhanced DOS can increase the SC gap and $T_c$. The same increasing ratio of $\Delta_1$ and $\Delta_2$ on type-I wrinkles clearly supports this scenario. On the other hand, once the strain is along the [100] direction (Fe-As direction), both the $d_{yz}$ and $d_{xz}$ band sink below $E_F$, leaving only the $d_{xy}$ band crossing $E_F$ (Fig. 4g). In this case, the loss of $d_{yz}$ Fermi surface leads to the disappearance of the $\Delta_1$ gap, and that only the $\Delta_2$ gap can be observed, yet the V-shape of the in-gap states is quite puzzling. This V-shape SC gap resembles the tunneling spectra in electron-doped LiFe$_{1-x}$Co$_x$As (ref. [20]), in which the $E_F$ only passes the $d_{xy}$ band possibly. We perform density functional theory (DFT) calculations and find the existence of compressive or tensile strain indeed has strong effect on shifting the local $d_{xz}$ band (Supplemental Fig. 8), which supports the band shifting scenario discussed above. The shifting of bands may induce the possible local

Lifshitz transition[32–35] at the wrinkle regions. We also note that the superconductivity in LiFeAs might relates to spin fluctuations. It is possible that the As anion height from Fe atoms[36] which will be affected by the local strain may modify the spin fluctuations and consequently tune the superconductivity. Recently, a nematicity order is also reported by STM[37] and ARPES[38] in LiFeAs. In iron-based superconductors, such as BaFe$_2$As$_2$ (ref. [8]) and Fe(Te,Se) (ref. [39]), a nematicity order emerges under strain. At the wrinkle locations of LiFeAs, the $C_4$ symmetry is broken, suggesting the existence of local strain which may also induce the nematicity order. Therefore, the nematicity order would be another possible origin of our observations. Nevertheless, we cannot rule out other scenarios of superconducting changing[40,41] and further theoretical understanding of the microscopic mechanism is required.

In summary, we have identified two types of wrinkles on LiFeAs surface. The orientation-dependent wrinkles accompanied with local uniaxial strains have very different influence on the superconductivity at the nanoscale, with one gets an enhancement while the other is suppressed. A possible Lifshitz transition scenario is proposed to explain these two distinct states of superconductivity in the vicinity of wrinkles on LiFeAs. Our observations suggest that the change of electronic structure induced by strain has strong influence for unconventional superconductivity in LiFeAs.

## Methods
**Single-crystal growth and scanning tunneling microscopy/spectroscopy (STM/S).** High-quality LiFeAs single crystals were synthesized by using the self-flux method[27]. LiFeAs crystals were mounted at a STM sample holder by epoxy in

a glove box and transferred to a ultra-high vacuum chamber where they were cleaved in-situ, and then immediately transferred to a STM scanner at 4.7 K. STM/S measurements were operated at 0.45 K by $^3$He single-shot technique (Unisoku). Polycrystal tungsten tips were etched chemically and calibrated on Au(111) surface before use. All STM images were acquired in the constant-current mode. The differential conductance ($dI/dV$) spectra and maps were obtained by a standard lock-in amplifier at a frequency of 973.0 Hz, with a modulation voltage of 0.2 mV. Magnetic fields were applied perpendicular to the samples.

**Noncontact atomic force microscopy (nc-AFM).** The nc-AFM measurements were conducted at liquid He temperature with a base ultra-high vacuum lower than $2 \times 10^{-10}$ mbar, where samples were cleaved in-situ. A commercial qPlus tuning fork sensor in frequency modulation mode with Pt/Ir tip was used to obtain the data. The resonance frequency was about 27.9 kHz and the stiffness was about 1800 N/m. The STM topography images were acquired in the constant-current mode. The constant-high and constant-force AFM modes were used to measure the real topography features of two types of wrinkles.

**Density functional theory (DFT) calculations.** DFT calculations employ the projector augmented wave (PAW) method encoded in the Vienna Ab initio Simulation Package (VASP)[42–44], and the generalized gradient approximation (GGA)[45] for the exchange-correlation functional is used. In order to simulate the tensile and compressive strain in the Fe-Fe direction and Fe-As direction respectively without causing a lot of calculations, we have established two different crystal models (Supplemental Fig. 8a, b). We create the supercells which has 120 atoms, including 40 Li atoms, 40 Fe atoms and 40 As atoms. In such a supercell, we stretch or compress one of the Fe-Fe bonds (Fe-As bonds), and then use this crystal model to calculate the orbital projected band structure. Although the angle between the two lattices is 45°, the x-axis of the Cartesian coordinate system is set along the Fe-Fe direction in both cases. The calculation parameters we used are as fellow: the plane-wave cut-off energy is 600 eV. In self-consistent calculations, k points are 13 × 1 × 8 with Γ centered. The energy convergence accuracy is $10^{-8}$ eV in the case of Fe-Fe while $10^{-6}$ eV in the case of Fe-As.

## Data availability

The data in this study are available from the corresponding authors on reasonable request.

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

## Acknowledgements

We thank H.M., F.Z., Y.-Y.Z., and G.Q. for helpful discussions. This work is supported by the Ministry of Science and Technology of China (2019YFA0308500, 2018YFA0305700, 2016YFA0401000), the National Natural Science Foundation of China (11888101, 52072401, 61888102, 51991340, 11820101003, 11921004, 11674371), and the Chinese Academy of Sciences (XDB28000000, XDB07000000).

## Author contributions

H.-J.G. and H.D. designed the experiments; L.C., W.L., and G.L. performed the STM experiment with the assistance of S.Z., L.K., and F.Y.; Q.Z., L.H., and X.L. performed the

AFM experiment; Y.W., K.J. performed the DFT calculations; G.D., X.W., and C.J. synthesized the LiFeAs single crystals; L.C., W.L., G.L., L.K., K.J., W.Z., J.H., H.D., and H.-J.G. analyzed the data with inputs from all other authors; L.C., W.L., G.L., and S.Z. plotted the figures. L.C., W.L., G.L., and K.J. wrote the manuscript with input from all other authors. H.-J.G. and H.D. supervised the project.

## Competing interests

The authors declare no competing interests.
