## [Peer Review File · Nature Communications]

REVIEWER COMMENTS

Reviewer #1 (Remarks to the Author):

In this paper the authors study the effect of naturally occurring wrinkles on superconductivity in LiFeAs. They find two types of wrinkles, one of which increases the superconducting gap and T_c , while the other shows a smaller gap with not much change in T_c . They attribute this to a strain induced band structure change.

The paper is logically written, the figures are clear, and the author's hypothesis for the reasons for the gap change seems reasonable. I just have a couple of comments/questions.

1) I would like the authors to have a deeper discussion on the different values of $2\Delta/k_B T_c$ they see for the different gaps. This is especially important since the type 2 region shows a small gap but the same T_c as the wrinkle free region. The big gap observed here is then a puzzle.

2) In plotting the gap with temperature, it is worth accounting for the temperature broadening of the gap by fitting the spectra to include temperature. That way, the small gap will not look like it is increasing with increasing temperature.

3) A small editorial comment: 'Inter-hole' in line 105 should be 'inner hole'.

Reviewer #2 (Remarks to the Author):

In this manuscript, Cao et al., present a STM study on the local strain effect on the surface of a Fe-based superconductor, LiFeAs. Previously, it has been demonstrated that the strain by using piezo can tune the global physical properties of Fe-based superconductors, including electronic nematicity in the orthorhombic phase and suppression of superconducting transition temperature. The authors find two types of wrinkles with different orientations on the surface of LiFeAs. They claim one enhances the local superconductivity while the other suppresses it. From the C_2 symmetric vortex and features in tunneling spectra, they further propose the Lifshitz transition due to the strain as the origin of the change in superconductivity. The novelty of this work, compared with the early studies on strained single crystals, is the reported local strain effect can either enhance or suppress high T_c superconductivity. The presented data and theoretical analysis are of high quality. Thus, I believe the results are of great interest and suitable for the readership of Nature Communications. However, I have several suggestions, questions and comments listed below that the authors should clarify before I can make the recommendation.

1. Possible lattice deformation due to cleavage: I agree it is not possible for STM to distinguish such a minute change in lattice structure if exists. The drift and the relaxation of the piezo as well as the different tip height can easily generate uncertainties in lateral scales of STM or AFM images. Thus, the estimated 0.4% of deformation in the supplementary is likely with large error bars. Nevertheless, this value is larger than the previous experiments with piezo and with different and larger effects (Nature Communications 9,2602 (2018)). Have the authors attempted to construct some model crystalline arrangement to simulate possible linear wrinkle along [110] or [100]? The DFT model in the supplementary seems to be a good starting point.

2. The authors rule out the bulk origin of structural defects and suggest the wrinkles are induced after cleavage. In this case, these wrinkles should exhibit random orientations and be easily found in larger field of view. While the authors do mention the wrinkle orientations are continuous (Line 72, 159, Fig

4c) and show the superconducting gap as the function of wrinkle orientation. However, only two different orientations are presented here: one along [110] and the other along [100]. The authors should present more images and tunneling spectrum with different orientation and gap size in the supplementary.

3. Line 204: The dip-hump features in superconducting tunneling spectrum are usually associated with electron-bosonic interactions; for example, PRL 109, 087002 or Nature Physics 11, 177. Do they also exhibit any enhancement or suppression at the wrinkles?

4. How is the gap size extracted from the data? The large coherence peaks can be easily identified at the low temperature. But the small gap Δ_2 is very difficult to see especially near T_c .

5. In Line 142, the authors state "Compared with its gap size, such T_c on type-II wrinkle is quite high". Why do the authors make such a statement? LiFeAs has multiple gaps. Which one did the author compare to? The BCS ratio is ~ 3.4 and doesn't seem to be too off.

6. In Line 143, "the bulk superconducting proximity effect while the intrinsic superconductivity on type-II wrinkle is already suppressed." Based on this argument, can the authors estimate the effective thickness of type II wrinkle?

7. Is the vortex at type II wrinkles also C2? Can the authors present a high resolution image in addition to Fig. S4d? The authors should also present the $dI/dV(r,E)$ map along two axes of the vortex to make this point stronger.

8. Recent STM and ARPES papers (PRB 100, 024506 & PRB 102, 184502) also report nematicity (fluctuation) in LiFeAs. Can this be also possible origin of the observations here (e.g. C2 nematicity stabilized at linear wrinkles)?

Minor issues:

9. Fig.2: It is difficult to tell where the boundary is. Please indicate their location and also label the distance for the waterfall plots.

10. Fig. S4: Please label the crystal orientations.

11. On the cleaved surface of other Fe-based compounds, Ref. 15 is one particular example. A review paper will be more desirable for the readers who are not familiar with this issue.

12. Typo: (black arrows) in Line 135.

13. Typo: When local strain is exists (Line 190)...."is" should be removed.

Reviewer #3 (Remarks to the Author):

This paper reports on local variation in superconductivity at wrinkles on cleaved LiFeAs; one of the Fe-based superconductors. Using scanning tunneling microscopy (STM) the authors observed superconducting gaps on wrinkles different from that on wrinkle-free area. They presume that the observed superconducting properties are related with the strain on the wrinkles, and with theoretical support from DFT calculations, they propose a scenario of strain-induced band shifts and Lifshitz transition to explain the observations.

The observation of the two types of peculiar superconducting states on wrinkles is solid and clear. Elongated vortices and abrupt type change with azimuthal angle of wrinkles are curious, but observations of these superconducting states only do not have strong impact enough to warrant publication in Nature Communications.

Then, the authors try to explain the superconducting behavior with band shift induced by strain and Lifshitz transition. Whereas the band shift of d_{xy} state was somehow observed experimentally in tunneling spectra (Fig. 4d) and was supported by theoretical calculations (Supple. Fig. 8), on d_{xz} and d_{yz} states, which are directly related to Lifshitz transition and therefore much more important, neither experimental observation nor theoretical support is provided. Basically there are no materials presented supporting Lifshitz transition experimentally or theoretically.

In fact, it is not clear whether the peculiar superconducting properties are really induced by strain. As

the authors mentioned, wrinkles have both tensile and compressive areas. No lattice expansion or shrinkage was detected in the STM images. I wonder whether only the top-most layer makes the wrinkles or not. If only the top layer pops up, monolayer region or at least region less interactive with the second layer is formed, which may exhibit different superconducting states from the bulk. Since the DFT calculations including strain does not seem supporting Lifshitz transition, the observed superconducting properties may not be simply explained with strain. Under these circumstances, I do not think the materials provided deserve publication in Nature Communications.

The results of DFT calculations presented in Supple. Fig. 8 seem strange to me. The results of no strain (d and g) should be same at least at Gamma point. But, they look quite different even at Gamma point. There must be some confusions/misunderstandings in the calculations.

Summary of the modifications

The revised parts are in red in the main text and supplementary notes. Following the constructive suggestions of the reviewers, we have revised the manuscript as below:

1. We add more discussions about the $2\Delta/k_B T_c$ values of different superconducting gaps (in supplementary notes 12). One table summarizing the result is added in supplementary note 12. We also add discussion about nematicity order as a possible origin of our observations (in the discussion part of main text).
2. We add more data including STM topography images as well as the tunneling spectra line-cut for type-I and type-II wrinkles with different orientations in supplementary notes 9 and 10.
3. Superconducting gaps are reevaluated following reviewer#1's suggestion, and Fig. 3d is replotted accordingly. The fitting method is described in supplementary note 11 following reviewer#2's suggestion.
4. Minor revisions have been done including adding the distance in Fig. 2, labeling the wrinkle boundary's locations in Fig. 2, and adding the orientations of crystal lattice in supplementary Fig. 4. We add the color-bar for every STM topography or map.
5. We replot supplementary Fig. 5 to show the transition from type-II to type-I wrinkle at the turn of wrinkles.
6. Four additional references in main text and two additional reference in supplementary materials are added following the reviewers' suggestions.
7. Typos and minor language issues are corrected as pointed out by the reviewers.

Point-by-point response to the comments from the reviewers

We thank the reviewers for their time and efforts for carefully reviewing our manuscript, and for providing valuable comments which help to improve the manuscript. We have addressed all comments in the following point-by-point reply and revised the manuscript accordingly. In this reply, comments from the reviewers are summarized in black italic typeface, our responses are in regular blue typeface, and our changes to the manuscript are in red.

Response to Reviewer #1:

In this paper the authors study the effect of naturally occurring wrinkles on superconductivity in LiFeAs. They find two types of wrinkles, one of which increases the superconducting gap and T_c , while the other shows a smaller gap with not much change in T_c . They attribute this to a strain induced band structure change.

The paper is logically written, the figures are clear, and the author's hypothesis for the reasons for the gap change seems reasonable. I just have a couple of comments/questions.

Response: We thank the reviewer for the positive remarks on the importance of our work. In the following, we provide our response to the reviewer's comments point-by-point.

Comment 1. *I would like the authors to have a deeper discussion on the different values of $2\Delta/k_B T_c$ they see for the different gaps. This is especially important since the type 2 region shows a small gap but the same T_c as the wrinkle free region. The big gap observed here is then a puzzle.*

Response: We thank the reviewer for raising this important issue. Following the suggestion, in supplementary materials, we add a table of the $2\Delta/k_B T_c$ for wrinkle-free, type-I and type-II wrinkle as shown below. We also add a discussion in supplementary note 12 as shown below.

	Wrinkle free		Type-I wrinkle		Type-II wrinkle
	Δ_1	Δ_2	Δ_1	Δ_2	Δ_2
Gap size	5.8	2.9	7.3	3.6	2.5
T_c	17		20.5		17
$2\Delta/k_B T_c$	7.9	4.0	8.3	4.1	3.4

Supplementary Table 1. Summary of gap size, T_c and $2\Delta/k_B T_c$.

“In the supplemental Table 1, we summarize the superconducting gap size, T_c and the ratio $2\Delta/k_B T_c$ for wrinkle-free, type-I and type-II wrinkles. The ratios are similar for wrinkle-free and type-I wrinkles, indicating that the pairing strength at type-I wrinkle region is close to that at the wrinkle-free region. Therefore, the increase of DOS at type-I wrinkle as proposed here is reasonable. On the other hand, Δ_2 at type-II wrinkle is small and the ratio at type-II wrinkle (~ 3.4) is close to the weak-coupling BCS regime, suggesting that the T_c at type-II wrinkle may be induced by the bulk superconducting proximity effect. Another possible explanation is that the d_{xz} as well as d_{yz} band sink below E_F , and only the smaller gap of d_{xy} due to the larger k value is observed (tunneling at the STM junction is insensitive to the M point), since the gap function in the FeSCs is found to be approximately proportional to $\cos(k_x)\cos(k_y)$ (P. Richard, T. Qian, H. Ding, Journal of Physics: Condensed Matter 27, 293203 (2015)). In this way, T_c is not driven by Δ_2 which is much smaller than the global superconducting pairing strength.”

Comment 2. *In plotting the gap with temperature, it is worth accounting for the temperature broadening of the gap by fitting the spectra to include temperature. That way, the small gap will not look like it is increasing with increasing temperature.*

Response: We thank the reviewer for this insightful suggestion. We modify our fitting method by considering the temperature broadening effect in the Fermi-Dirac distribution function, and convolve it with the Dynes formula (see details in supplementary note 11). The results are replotted in Fig. 3d where the intrinsic gap-temperature relations are displayed. The temperature broadening effect is also considered in fitting the gaps of type-I wrinkle and wrinkle-free regions.

Fig. R1. Replot of the Fig. 3d in main text.

Comment 3. *A small editorial comment: ‘Inter-hole’ in line 105 should be ‘inner hole’.*

Response: We thank the reviewer to point out this typo. We have corrected it in the manuscript.

Response to Reviewer #2

In this manuscript, Cao et al., present a STM study on the local strain effect on the surface of a Fe-based superconductor, LiFeAs. Previously, it has been demonstrated that the strain by using piezo can tune the global physical properties of Fe-based superconductors, including electronic nematicity in the orthorhombic phase and suppression of superconducting transition temperature. The authors find two types of wrinkles with different orientations on the surface of LiFeAs. They claim one enhances the local superconductivity while the other suppresses it. From the C2 symmetric vortex and features in tunneling spectra, they further propose the Lifshitz transition due to the strain as the origin of the change in superconductivity. The novelty of this work, compared with the early studies on strained single crystals, is the reported local strain effect can either enhance or suppress high T_c superconductivity. The presented data and theoretical analysis are of high quality. Thus, I believe the results are of great interest and suitable for the readership of Nature Communications.

However, I have several suggestions, questions and comments listed below that the authors should clarify before I can make the recommendation.

Response: We thank the reviewer for the high evaluation of the novelty and importance of our work. We response to these comments in the following point-by-point reply.

Comment 1. *Possible lattice deformation due to cleavage: I agree it is not possible for STM to distinguish such a minute change in lattice structure if exists. The drift and the relaxation of the piezo as well as the different tip height can easily generate uncertainties in lateral scales of STM or AFM images. Thus, the estimated 0.4% of deformation in the supplementary is likely with large error bars. Nevertheless, this value is larger than the previous experiments with piezo and with different and larger effects (Nature Communications 9,2602 (2018)). Have the authors attempted to construct some model crystalline arrangement to simulate possible linear wrinkle along [110] or [100]? The DFT model in the supplementary seems to be a good starting point.*

Response: We agree with the reviewer that the estimated 0.4% lattice deformation has large error bars, and we have added the following statement in the supplementary note 1:

“We note that the value of 0.4 % is with large error bar since the effects such as drifting or piezo calibration could contribute to this small value.”

According to the STM images, the width of a wrinkle is approximately 10 nm, which means that a wrinkle contains dozens of unit cells. We note that it is quite difficult to simulate the real wrinkle in the DFT slab model because of the huge expense on calculation resources. In addition, we speculate that the local strain on wrinkle has a complex distribution (Supplementary Fig. 3b) and it is unrealistic to fully construct such a model. Therefore, we employ the simple model to qualitatively capture the

physics of band shift under strain. We also agree with the reviewer that simulating the real linear wrinkle is important to uncover the microscopic effect of local strain on the local electronic states in LiFeAs. We hope that our work will stimulate future work to construct a more realistic slab model to investigate the strain effect more thoroughly.

Comment 2. *The authors rule out the bulk origin of structural defects and suggest the wrinkles are induced after cleavage. In this case, these wrinkles should exhibit random orientations and be easily found in larger field of view. While the authors do mention the wrinkle orientations are continuous (Line 72, 159, Fig 4c) and show the superconducting gap as the function of wrinkle orientation. However, only two different orientations are presented here: one along [110] and the other along [100]. The authors should present more images and tunneling spectrum with different orientation and gap size in the supplementary.*

Response: We thank the reviewer for this suggestion. In our experiment, the linear wrinkles easily extend a few hundreds of nm in length, and as a result, we normally image fraction of these wrinkles. The wrinkles do make turns at certain points and we occasionally capture those features. One example is shown in supplementary note 5. Although it is difficult to observe different wrinkle orientations in the same field of view, we do observe different wrinkle orientations on different locations of the same single crystalline sample surface. We add more STM topography and tunneling spectra with different orientations and gap size in the supplementary materials (see supplementary note 9 and 10), as shown below.

Fig. R2 (Supplementary Figure 9). More data for type-I wrinkles. a-d, STM topography images for type-I wrinkles along different orientations. The angle between the wrinkles' orientations and the [110]

crystal direction are marked in the upper left corner. **e-h**, The corresponding dI/dV spectra line-cut taken along the solid arrows in **a-d**, respectively. Enhanced gap emerges at the location of type-I wrinkle. Black arrows indicate the wrinkle boundary.

Fig. R2 (Supplementary Figure 10). More data for type-II wrinkles. **a-d**, STM topography images for type-II wrinkles along different orientations. The angle between the wrinkles' orientations and the $[110]$ crystal direction are marked in the upper left corner. **e-h**, The corresponding dI/dV spectra line-cut taken along the solid arrows in **a-d**, respectively. Suppressed gap emerges at the location of type-II wrinkle. Black arrows indicate the wrinkle boundary.

Comment 3. Line 204: *The dip-hump features in superconducting tunneling spectrum are usually associated with electron-bosonic interactions; for example, PRL 109, 087002 or Nature Physics 11, 177. Do they also exhibit any enhancement or suppression at the wrinkles?*

Response: We zoom in Fig. 4d as shown below. The dip-hump features are marked by the black, red and blue arrows for wrinkle-free, type-I and type-II region, respectively. Compare with the wrinkle-free region, the dip-hump feature at type-I region have no obvious enhancement or suppression, while the dip-hump feature at type-II region exhibits some suppression. However, we note that LiFeAs is a multi-band Fe-based superconductor, and the corresponding multi-band and multi-gap feature can lead to the dip-hump feature.

Fig. R3. Zoom-in of Fig. 4d.

Comment 4. *How is the gap size extracted from the data? The large coherence peaks can be easily identified at the low temperature. But the small gap Δ_2 is very difficult to see especially near T_c .*

Response: We use the Dynes formula to extract the gap sizes (Shan, L. *et al.* Nat. Phys. 7, 325-331 (2011)). To clarify this point, we add a description in supplementary note 11:

“The superconducting gaps are extracted by fitting the dI/dV spectra. First, the dI/dV below T_c are normalized by dividing the normal state dI/dV ($T > T_c$) to remove the background. Next, we fit the normalized dI/dV curve using the Dynes formula as following:

$$\frac{dI}{dV} \propto \int \int_0^{2\pi} \frac{df(E + eV, T)}{dV} N(E) d\theta_k dE$$

where

$$N(E) = \text{Re} \left(\frac{E - i\Gamma}{\sqrt{(E - i\Gamma)^2 - \Delta(\theta_k)^2}} \right)$$

and the $f(E + eV, T)$ is the Fermi-Dirac distribution function at the temperature T . E is the energy, Γ the inverse of quasiparticle lifetime, and $\Delta(\theta_k)$ the gap function in k -space. The temperature broadening effect is included in the Fermi-Dirac distribution function. According to the ARPES results, the $\Delta(\theta_k)$ for LiFeAs exhibits a 4-fold symmetry. So we adopt the 4-fold symmetry gap function for the gaps. As previously reported (Shan, L. *et al.* Nat. Phys. 7, 325-331 (2011)), in the fitting of gaps of the type-I wrinkle region and wrinkle-free region, the total dI/dV spectra value is contributed by the two gaps Δ_1 and Δ_2 : $dI/dV = P dI/dV(\Delta_1) + (1-P) dI/dV(\Delta_2)$. P is the weight and we adopt $P = 0.7$ in our fitting. And $\Delta_1 = \Delta_a(1 + 0.1\cos(4\theta_k))$, $\Delta_2 = \Delta_b(1 + 0.1\cos(4\theta_k))$. In the fitting of gap of the type-II wrinkle region, only Δ_2 are used and $\Delta_2 = \Delta_b(0.1 + \cos(4\theta_k))$.”

Below, we show three examples of the fitting result for type-I wrinkle, wrinkle-free and type-II wrinkle region. The fitting curves are consistent with the experimental data, demonstrating the validity of our fitting method.

Fig. R4. Examples of the superconducting gap fitting results.

Comment 5. In Line 142, the authors state “Compared with its gap size, such T_c on type-II wrinkle is quite high”. Why do the authors make such a statement? LiFeAs has multiple gaps. Which one did the author compare to? The BCS ratio is ~ 3.4 and doesn't seem to be too off.

Response: We realize that the statement in the original manuscript is somewhat misleading. Therefore, we delete this statement in the revised manuscript. In the supplementary note 12, we add a table of the $2\Delta/k_B T_c$ for the wrinkle-free, type-I and type-II wrinkle as shown below. We also add a discussion in supplementary note 12 as shown below.

	Wrinkle free		Type-I wrinkle		Type-II wrinkle
	Δ_1	Δ_2	Δ_1	Δ_2	Δ_2
Gap size	5.8	2.9	7.3	3.6	2.5
T_c	17		20.5		17
$2\Delta/k_B T_c$	7.9	4.0	8.3	4.1	3.4

Supplementary Table 1. Summary of gap size, T_c and $2\Delta/k_B T_c$.

“In the supplemental Table 1, we summarize the superconducting gap size, T_c and the ratio $2\Delta/k_B T_c$ for wrinkle-free, type-I and type-II wrinkles. The ratios are similar for wrinkle-free and type-I wrinkles, indicating that the pairing strength at type-I wrinkle region is close to that at the wrinkle-free region. Therefore, the increase of DOS at type-I wrinkle as proposed here is reasonable. On the other hand, Δ_2 at type-II wrinkle is small and the ratio at type-II wrinkle (~ 3.4) is close to the weak-coupling BCS regime, suggesting that the T_c at type-II wrinkle may be induced by the bulk superconducting proximity effect. Another possible explanation is that the d_{xz} as well as d_{yz} band sink below E_F , and only the smaller gap of d_{xy} due to the larger k value is observed (tunneling at the STM junction is insensitive to the M point), since the gap function in the FeSCs

is found to be approximately proportional to $\cos(k_x)\cos(k_y)$ (P. Richard, T. Qian, H. Ding, *Journal of Physics: Condensed Matter* 27, 293203 (2015)). In this way, T_c is not driven by Δ_2 which is much smaller than the global superconducting pairing strength.”

Comment 6. *In Line 143, “the bulk superconducting proximity effect while the intrinsic superconductivity on type-II wrinkle is already suppressed.” Based on this argument, can the authors estimate the effective thickness of type II wrinkle?*

Response: The superconducting proximity effect is within the coherence length. For LiFeAs, a previous result shows that the coherence length along the a or b axis is 4.64 nm while along the c axis is 1.50 nm (*J. Phys. Soc. Jpn.* 80 (2011) 013706). This would give an estimated effective thickness of type II wrinkle of ~ 1.5 nm. However, we also note that the proximity effect on type-II wrinkle might come from the horizontal plane instead of from the vertical layers lying beneath. In such a case, it is difficult to estimate the effective thickness.

Comment 7. *Is the vortex at type II wrinkles also C_2 ? Can the authors present a high resolution image in addition to Fig. S4d? The authors should also present the $dI/dV(r,E)$ map along two axes of the vortex to make this point stronger.*

Response: The vortices at type-II wrinkles are also C_2 . We note that the two dI/dV maps in supplementary Fig. S4 are of similar resolution. This can be seen from the similar four-point-star shaped vortices at the wrinkle-free regions in Fig. S4b and S4d. The reason that Fig. S4d looks more blurred than Fig. S4b lies in the fact that the SC gap at type-II wrinkle is not a hard gap and owns a finite density of states (DOS) at the Fermi level (Fig. 1f). The non-zero DOS fills up the spacings between adjacent vortices, making it difficult to resolve the boundary of a single vortex. The linecut measurements would have the same issue. If we check carefully at the two vortices on type-II wrinkle at the bottom right corner of Fig. S4d, the C_2 symmetry of vortices on the wrinkle can be identified.

Comment 8. *Recent STM and ARPES papers (PRB 100, 024506 & PRB 102, 184502) also report nematicity (fluctuation) in LiFeAs. Can this be also possible origin of the observations here (e.g. C_2 nematicity stabilized at linear wrinkles)?*

Response: We thank the reviewer for bringing up this point. In Fe-based superconductors such as BaFe_2As_2 and $\text{Fe}(\text{Te},\text{Se})$, a nematicity order (or fluctuations) is observed to emerge under the strain. At the wrinkle locations of LiFeAs, the C_4 symmetry breaking would induce the possible nematicity order of fluctuation. Therefore, the nematicity (or its fluctuations) is a possible origin of our observations. We add the discussion of the nematicity (or its fluctuations) in the discussion part in the revised manuscript and cite the two references, as shown below:

“Recently, a nematicity order is also reported by STM (*Phys. Rev. B* 100, 024506

(2019)) and ARPES (Phys. Rev. B 102, 184502 (2020)) in LiFeAs. In Fe-based superconductors such as BaFe₂As₂ (Science 345, 657-660 (2014)) and Fe(Te,Se) (Nat. Phys. <https://doi.org/10.1038/s41567-021-01254-8> (2021)), a nematicity order emerges under strain. At the wrinkle locations of LiFeAs, the C₄ symmetry is broken, suggesting the existence of local strain which may also induce the nematicity order. Therefore, the nematicity order would be another possible origin of our observations.”

Minor issues:

Comment 9. *Fig.2: It is difficult to tell where the boundary is. Please indicate their location and also label the distance for the waterfall plots.*

Response: We adopt the reviewer’s suggestion. We indicate the boundary location and label the distance for the waterfall plots in Fig. 2 in the revised manuscript.

Comment 10. *Fig. S4: Please label the crystal orientations.*

Response: We label the crystal orientations in supplementary Fig. 4 in the revised manuscript.

Comment 11. *On the cleaved surface of other Fe-based compounds, Ref. 15 is one particular example. A review paper will be more desirable for the readers who are not familiar with this issue.*

Response: We adopt the reviewer’s suggestion and add a review paper (Ref. 16: Hoffman, J. E. Spectroscopic scanning tunneling microscopy insights into Fe-based superconductors. Rep. Prog. Phys. 74, 124513 (2011).) which contain this issue in Fe-based compounds.

Comment 12. *Typo: (black arrows) in Line 135.*

Response: We correct the related contents in the revised manuscript.

Comment 13. *Typo: When local strain is exists (Line 190).... ”is” should be removed.*

Response: We correct the related contents in the revised manuscript.

Response to Reviewer #3

This paper reports on local variation in superconductivity at wrinkles on cleaved LiFeAs; one of the Fe-based superconductors. Using scanning tunneling microscopy (STM) the authors observed superconducting gaps on wrinkles different from that on wrinkle-free area. They presume that the observed superconducting properties are related with the strain on the wrinkles, and with theoretical support from DFT calculations, they propose a scenario of strain-induced band shifts and Lifshitz transition to explain the observations.

The observation of the two types of peculiar superconducting states on wrinkles is solid and clear. Elongated vortices and abrupt type change with azimuthal angle of wrinkles are curious, but observations of these superconducting states only do not have strong impact enough to warrant publication in Nature Communications.

Response: We thank the review's positive comment on our experimental observations. Here, we want to point out that our result is the first observation that the superconductivity is clearly modified by local strains in a Fe-based superconductor. Compared with the previous hydrostatic pressure experiment results in LiFeAs where the pressure is applied in all directions and the T_c is always suppressed, we show the bi-directional superconductivity changing behavior where the strain in different orientations can either enhance or suppress the SC gaps. More surprisingly, the switch from enhancement to suppression is discontinuous, suggesting the Lifshitz transition of Fermi surface topology may play an important role in the superconducting pairing strength.

Then, the authors try to explain the superconducting behavior with band shift induced by strain and Lifshitz transition. Whereas the band shift of d_{xy} state was somehow observed experimentally in tunneling spectra (Fig. 4d) and was supported by theoretical calculations (Supple. Fig. 8), on d_{xz} and d_{yz} states, which are directly related to Lifshitz transition and therefore much more important, neither experimental observation nor theoretical support is provided. Basically there are no materials presented supporting Lifshitz transition experimentally or theoretically.

Response: Actually, from the dI/dV spectra, we show the evidence for the d_{xz}/d_{yz} bands shift in our manuscript (line 167-168, 172-176 in original version, or line 167-168, 172-177 in this revised version). The hump feature around the Fermi level suggests the band top of d_{xz} , which is below the Fermi level at the wrinkle-free region and above the Fermi level at type-I wrinkle. So, it indicates the band shift of d_{xz} in the sketches shown in Fig. 4e and 4f. In addition, the theoretical calculations in Supplementary Fig. 8 indeed demonstrate the shift of d_{xz} under a local strain effect (see the arrows marked in Supplementary Fig. 8 c-h). Based on the discussion above, we believe we have provided experimental observations and theoretical support of the band shift of d_{xz} .

In fact, it is not clear whether the peculiar superconducting properties are really induced by strain. As the authors mentioned, wrinkles have both tensile and compressive areas. No lattice expansion or shrinkage was detected in the STM images. I wonder whether only the top-most layer makes the wrinkles or not. If only the top layer pops up, monolayer region or at least region less interactive with the second layer is formed, which may exhibit different superconducting states from the bulk. Since the DFT calculations including strain does not seem supporting Lifshitz transition, the observed superconducting properties may not be simply explained with strain.

Response: Although the accurate lattice distortion cannot be detected by STM, from the vortex structure, we can infer the strain effect: the vortex structure change from C_4 symmetry in wrinkle-free region to C_2 symmetry in the wrinkle region (as we analyzed in manuscript line 146-154 in original version, or line 145-153 in this revised version). We also note that application of external strain in LiFeAs can change the vortex structure. Besides, the DFT calculations also give a support to the Lifshitz transition since the band top of d_{xz} shifts under the local strain (arrows in Supplementary Fig. 8c-h).

We cannot evaluate the how many layers do the wrinkles possess because the STM is only sensitive to the surface. However, from the orientation-dependent effects of wrinkles, one can deduce that the strains in different orientations have distinct effects on the superconducting gaps. The orientation-dependent effects cannot be simply explained by the less interactive between the pops-up region and the underneath region.

Under these circumstances, I do not think the materials provided deserve publication in Nature Communications.

The results of DFT calculations presented in Supple. Fig. 8 seem strange to me. The results of no strain (d and g) should be same at least at Gamma point. But, they look quite different even at Gamma point. There must be some confusions/misunderstandings in the calculations.

Response: As we displayed in supplementary Fig. 8a and 8b, the DFT band structures in supplementary Fig. 8d and 8g are based on the different slab models and along the different high-symmetry directions. In supplementary Fig. 8d, the periodic direction is along the Fe-Fe direction (marked by the black arrow in supplementary Fig. 8a or Fig. R5a). Thus, the band structure which is perpendicular to this direction must generate the projection along this direction (Fig. R5a). In supplementary Fig. 8g, the periodic direction is along one of the Fe-As direction (marked by the black arrow in supplementary Fig. 8b or Fig. R5b). Thus, the band structures for these two slab models should be different.

Fig. R5. Sketch of the band projection in the DFT model. **a**, Sketch of the Fermi surface of LiFeAs in 1-Fe Brillouin zone. The solid red and purple circles are for the hole Fermi surface pockets and the solid blue circles are for the electron Fermi surface pockets. The black arrow indicates the periodic direction (k_1) of the slab model in supplementary Fig. 8a, which is also the dispersion direction in supplementary Fig. 8c-e. Therefore, the band structure along the direction perpendicular to k_1 must generate projection along the k_1 direction, as shown by the dashed circles. **b**, Same of **a**, but correspond to the slab model in supplementary Fig. 8b, and the periodic direction is k_2 .

REVIEWERS' COMMENTS

Reviewer #1 (Remarks to the Author):

The authors have addressed most of my concerns. I would strongly suggest that the discussion on the gap size and its connection to T_c be moved to the main text. Other than that I am happy to recommend publication.

Reviewer #3 (Remarks to the Author):

The authors gave me replies to my comments on the shift of the electronic states and the Lifshitz transition. Some of them are, however, not really satisfactory yet. Actually, by changing the lattice constant, the energy level of the electronic states shifts. It is quite natural and trivial; without complicated calculations presented in Fig. S8, shifts in the energy level are expected. Important things related to the Lifshitz transition is whether the top or bottom of the electronic bands crosses the Fermi level or not. The calculated results presented in Fig. S8 fail to demonstrate the Fermi level crossing by strain.

Then, moving to the experimental side, the authors observed the shift of bump features across the Fermi level between two spectra shown in Figs. 3a and 3b, but this is between the wrinkle-free site and type-I wrinkle site, where no significant changes are found in their superconductivities other than the gap amplitude; no Lifshitz transition there.

On the other hand, between the wrinkle-free site and type-II wrinkle site, where the authors expect the Lifshitz transition, no shift in the electronic states across the Fermi level is observed. In Fig. S6, the three spectra showing a shift in the bump feature are presented, but only far above the Fermi level.

I thus reiterate my previous statement "Basically there are not materials presented supporting Lifshitz transition experimentally and theoretically".

The relevant discussion should be improved.

Point-by-point response to the comments from the reviewers

We thank the reviewers again for their constructive reviewing process, which helps to improve our manuscript and strengthen the statements. We have addressed all comments in the following point-by-point reply and revised the manuscript accordingly. In this reply, comments from the reviewers are summarized in black italic typeface, our responses are in regular blue typeface, and our changes to the manuscript are in red.

Reviewer #1 (Remarks to the Author):

The authors have addressed most of my concerns. I would strongly suggest that the discussion on the gap size and its connection to T_c be moved to the main text. Other than that I am happy to recommend publication.

Response: We highly appreciate the reviewer's suggestion and move the discussion about the gap size and T_c to the discussion part of main text (on page 5-6, line 190-201):

“In the Supplemental Table 1, we summarize the superconducting gap size, T_c and the ratio $2\Delta/k_B T_c$ for wrinkle-free, type-I and type-II wrinkles. The ratios are similar for wrinkle-free and type-I wrinkles, indicating that the pairing strength at type-I wrinkle region is close to that at the wrinkle-free region. Therefore, the increase of DOS at type-I wrinkle as proposed here is reasonable. On the other hand, Δ_2 at type-II wrinkle is small and the ratio at type-II wrinkle (~ 3.4) is close to the weak-coupling BCS regime, suggesting that the T_c at type-II wrinkle may be induced by the bulk superconducting proximity effect. Another possible explanation is that the d_{xz} as well as d_{yz} bands sink below E_F , and only the smaller gap of d_{xy} is observed (tunneling at the STM junction is insensitive to the M point), since the gap function in the FeSCs is found to be approximately proportional to $\cos(k_x)\cos(k_y)$ ³¹. In this way, T_c is not driven by Δ_2 which is much smaller than the global superconducting pairing strength.”

Reviewer #3 (Remarks to the Author):

The authors gave me replies to my comments on the shift of the electronic states and the Lifshitz transition. Some of them are, however, not really satisfactory yet.

Actually, by changing the lattice constant, the energy level of the electronic states shifts. It is quite natural and trivial; without complicated calculations presented in Fig. S8, shifts in the energy level are expected. Important things related to the Lifshitz transition is whether the top or bottom of the electronic bands crosses the Fermi level or not. The calculated results presented in Fig. S8 fail to demonstrate the Fermi level crossing by strain.

Then, moving to the experimental side, the authors observed the shift of bump features across the Fermi level between two spectra shown in Figs. 3a and 3b, but this is

between the wrinkle-free site and type-I wrinkle site, where no significant changes are found in their superconductivities other than the gap amplitude; no Lifshitz transition there.

On the other hand, between the wrinkle-free site and type-II wrinkle site, where the authors expect the Lifshitz transition, no shift in the electronic states across the Fermi level is observed. In Fig. S6, the three spectra showing a shift in the bump feature are presented, but only far above the Fermi level.

I thus reiterate my previous statement “Basically there are not materials presented supporting Lifshitz transition experimentally and theoretically”.

The relevant discussion should be improved.

Response: We thank the review’s comment about the Lifshitz transition. In order to soften the statement of Lifshitz transition, in the revised version, we replace some of the statement of “Lifshitz transition” by “band shifting”, which is directly supported by the experimental results in Figs. 3a and 3b and Fig. S6. We state Lifshitz transition as a possible consequence of the band shifting. The modifications are summarized as below:

On page 2, line 36, Abstract: “suggesting that **the band shifting** induced by directional pressure may”

On page 6, line 204: “local strain on wrinkle causes the **shifting of bands around Γ point.**”

On page 6, line 212: “the **loss** of d_{yz} Fermi surface”

On page 6, line 218-219: “which supports the **band shifting** scenario discussed above. **The shifting of bands may induce the possible local Lifshitz transition³²⁻³⁵ at the wrinkle regions.**”

On page 6, line 234: “A **possible Lifshitz transition** scenario is”

On page 15, line 441: “**Fig. 4 Uniaxial strain induced band shifting.**”